# Workout Classification Using a Convolutional Neural Network in Ensemble Learning

**DOI:** 10.3390/s24103133

**Published:** 2024-05-15

**Authors:** Gi-Seung Bang, Seung-Bo Park

**Affiliations:** Department of Software Convergence Engineering, Inha University, Incheon 22212, Republic of Korea; rltmdqkd12@naver.com

**Keywords:** ensemble learning, convolutional neural network, MediaPipe, computer vision, home workout

## Abstract

To meet the increased demand for home workouts owing to the COVID-19 pandemic, this study proposes a new approach to real-time exercise posture classification based on the convolutional neural network (CNN) in an ensemble learning system. By utilizing MediaPipe, the proposed system extracts the joint coordinates and angles of the human body, which the CNN uses to learn the complex patterns of various exercises. Additionally, this new approach enhances classification performance by combining predictions from multiple image frames using an ensemble learning method. Infinity AI’s Fitness Basic Dataset is employed for validation, and the experiments demonstrate high accuracy in classifying exercises such as arm raises, squats, and overhead presses. The proposed model demonstrated its ability to effectively classify exercise postures in real time, achieving high rates in accuracy (92.12%), precision (91.62%), recall (91.64%), and F1 score (91.58%). This indicates its potential application in personalized fitness recommendations and physical therapy services, showcasing the possibility for beneficial use in these fields.

## 1. Introduction

In recent years, social distancing due to the spread of COVID-19 has become the norm, leading to a surge in non-face-to-face activities. Facilities like gyms are especially vulnerable to spreading disease, which paved the way for home training to gain popularity as an alternative. Figure 1 shows the trend in Web searches by using home training terms on Google. It is evident from Figure 1 that there was a partial increase in searches for home training during the peak of the pandemic. However, even after it subsided for a while, the interest in home training increased again.

According to the Big Data Team of the Consumer Information and Education Department at the Korea Consumer Agency, the trend in home training indicates increased public interest due to the pandemic. The proportion of consumers investing in home training is also high. There is an anticipated increase not only in the demand for equipment but also for home training platforms [1].

At this juncture, the rapid development of artificial intelligence and computer vision technology has spurred advancements in the field of real-time human action recognition. In particular, open-source software frameworks like MediaPipe [2] have been leading innovations in this area by providing the ability to detect and track individual body part positions in video streams in real time.

Also, recent studies have emphasized the potential of convolutional neural networks (CNNs) and ensemble learning methods to enhance exercise classification accuracy, showcasing significant advancements in remote fitness coaching and rehabilitation.

The advancements in these technologies have offered new opportunities in the home training market, and the research and development of systems and models related to exercise correction are being conducted. However, the current popular systems have difficulty accurately identifying the exercises performed. Most users must manually input the type of exercise after completion, which hinders convenience and impedes accurate analysis and correction.

To address these challenges, we present an advanced classification model that utilizes MediaPipe for accurate joint coordinate and angle extraction, integrating these features into a CNN for exercise classification. Our model’s accuracy is further improved through an ensemble method optimizing frame-by-frame classification results.

Through this, the system can classify and analyze exercises on its own, allowing users to correct their posture without additional input. Furthermore, our results are expected to contribute to the improvement of in-home training system usability and, more broadly, to the enhancement of personal health management and exercise efficiency.

## 2. Exercise Classification Methods

### 2.1. Technical Background

#### 2.1.1. MediaPipe

MediaPipe is an open-source software framework developed by Google and capable of performing real-time body posture estimation, face recognition, and hand gesture recognition from videos. This technology has been utilized in numerous research projects and has established itself as a vital tool for analyzing human behavior and activities. MediaPipe allows the extraction and use of key body parts from the coordinates shown in Figure 2. In this research, we use this technology to extract body posture coordinates and calculate joint angles.

#### 2.1.2. CNN

Convolutional neural networks have become a cornerstone in the field of deep learning for visual image processing and analysis. CNNs are designed to automatically learn hierarchical spatial features from input images, exhibiting exceptional performance in image recognition, classification, and object detection tasks [3].

The architecture of a CNN typically comprises multiple convolutional layers. These layers apply convolutional filters to the input, capturing spatial features such as edges or shapes. After the convolutional layers are pooling layers, which reduce the dimensions of the data and compress information, thereby decreasing computational load. After passing through several convolutional and pooling layers, the higher-order reasoning of the network culminates in an output layer that provides classification results.

In exercise classification, CNNs can process video frames as a sequence of images, extracting essential features for identifying various exercises. By training on datasets encompassing a wide range of exercises performed by various individuals, CNNs can generalize new and unseen examples quite well, making them suitable models for exercise classification systems.

In this study, we apply a CNN to capture complex human movement patterns and provide comprehensive analysis for each type of exercise.

#### 2.1.3. The Ensemble Approach

Ensemble learning is a methodology that combines multiple classifiers to achieve better performance. Soft voting (a technique within ensemble learning) probabilistically averages the predictions of each classifier to make a final decision. This method allows more robust and stable predictions through a voting process among classifiers. The effectiveness of soft voting has been proven in various classification problems, especially with data that are noisy or that have high variability. In this study, we utilize an algorithm with a configuration similar to soft voting because we are working with data prone to outliers [4].

#### 2.1.4. InfiniteRep

In this study, we utilize the Fitness Basic Dataset by Infinity AI. This dataset is designed to be easily accessible to engineers building foundational products in the Wingate fitness application domain [5]. Also known as InfiniteRep, the dataset comprises videos of avatars performing typical exercises. Figure 3 presents a scene from an InfiniteRep video, capturing an avatar in the midst of a representative exercise.

The decision to utilize the Fitness Basic Dataset by Infinity AI was predicated on a multifaceted evaluation of its suitability for advancing the objectives of this research. This dataset, characterized by its extensive collection of avatar-based exercise videos, was deemed instrumental for the CNN model’s learning process. It encompasses a wide spectrum of exercises, providing a robust foundation for the model to learn and classify exercise types with high accuracy.

Moreover, the dataset’s avatar-based approach circumvents the ethical and logistical complexities inherent in the collection of real human exercise data. This consideration was paramount, given the increasing scrutiny and regulatory constraints on research involving human subjects. By adopting a virtual dataset, this study aligns with ethical research practices, ensuring that the integrity of the research process is maintained.

Furthermore, the choice to employ virtual data over real data was driven by the imperative of research reproducibility and consistency. In the context of exercise classification, where data diversity and quality are critical, the Fitness Basic Dataset ensures a controlled environment free from the variability and noise that are typical in a real-world data collection. This controlled dataset enables a more focused analysis of the model’s performance, devoid of extraneous variables that could compromise the validity of the research.

The selection of the Fitness Basic Dataset thus reflects a deliberate strategy to balance the need for comprehensive and diverse exercise data with ethical, legal, and methodological considerations when conducting high-caliber research in real-time exercise classification.

### 2.2. Literature Review

#### 2.2.1. Exercise Posture Correction Systems (MediaPipe)

Recently, the demand for home training has increased owing to COVID-19, raising interest in posture estimation and correction. Some studies have used MediaPipe for this purpose but encountered reduced recognition rates due to certain anomalies, which they addressed using voting algorithms. However, these studies have limitations because they only classify exercises such as squats, lunges, and dumbbell curls, leaving similar exercises unverified [6,7].

#### 2.2.2. Yoga and Pilates Pose Classification

Predominant in previous exercise classification models are studies on yoga and pilates. These studies primarily utilized multilayer perceptron (MLP) for classification and included models that use transfer learning with pre-trained models like MobileNet or Inception [8,9]. Some papers employed models using recurrent neural networks like long short-term memory (LSTM) [10]. While most boast accuracy rates above 95%, one limitation is that the rates are based on analyses of individual photo frames rather than continuous motion videos. This study classifies continuous motion by using CNN models employed in prior research and by experimenting with ensemble learning and LSTM.

#### 2.2.3. Using CNNs to Classify Exercise Videos

The application of CNNs to classifying exercise videos has emerged as a significant advancement in machine learning and physical therapy. Studies have demonstrated the effectiveness of CNNs and related technologies in enhancing the precision of exercise classification, thus providing novel insights into the potential of machine learning to support health and fitness.

A pioneering study by Arrowsmith et al. explored the feasibility of using single-camera systems paired with CNNs for automated monitoring of physiotherapy exercises, targeting conditions such as low-back and shoulder pain. This work emphasized the robustness of CNN models against variations in filming angles and highlighted the potential of smartphone cameras and machine learning in tracking adherence to physiotherapy programs [11].

Complementing this, research by another group in 2024 focused on synchronizing and preprocessing sensor data for machine learning classifiers, showcasing the importance of data quality in achieving accurate exercise classification. Their findings contribute to the understanding of challenges and solutions in using wearable sensors and machine learning for human activity recognition [12].

Additionally, recent advancements highlighted in a 2020 study showcased the effective use of deep CNNs for rehabilitation exercise recognition and evaluation. Utilizing smart sensors, that study explored a deep-learning framework that significantly enhances the interpretation of complex sensor data for activity and exercise recognition. This approach not only demonstrates the efficacy of CNNs in processing sensor-derived data but sets a new precedent for the application of deep learning in physical rehabilitation, paving the way for innovative developments in this field [13].

These studies collectively underscore the growing importance of CNN and machine learning technologies in advancing the field of exercise classification. The meaningful results demonstrated by utilizing CNNs for video classification highlight the transformative potential of these technologies in analyzing complex motion data, enabling more accurate and efficient monitoring and classification of physical activities. This body of work not only enhances our understanding of the capabilities of CNNs in the realm of digital health but lays the foundation for future innovations in personalized fitness and rehabilitation programs.

Our research leverages the advancements in CNN technology to further explore and refine the classification of exercise videos. By doing so, we contribute to the evolving landscape of machine-learning applications in health and fitness, seeking to improve patient engagement, treatment outcomes, and the accessibility of home-based therapeutic interventions. Through the application of CNNs, we anticipate uncovering new insights and methodologies that will advance the effectiveness of remote physiotherapy monitoring and exercise recognition, aligning with the pioneering efforts highlighted in the aforementioned studies.

### 2.3. The System Architecture

The overall concept for the system we implement is shown in Figure 4. The proposed algorithm consists of joint data preprocessing, model training (fit), and exercise classification (prediction). First, we extract body joint data from pre-classified exercise video data, train the model with these data, and then classify exercises in real time based on the trained model.

### 2.4. Classification Methods

#### 2.4.1. Media Data Preprocessing

We use Open Computer Vision (OpenCV) to load videos and utilize the MediaPipe framework to extract 22 types of body data, excluding facial data. MediaPipe provides coordinate values (joint points on *x*, *y*, and *z* planes) for a total of 66 features.

#### 2.4.2. Extract Data Features

Most exercises require body rotation, and utilizing various joint angles can aid in recognition and classification. Using the coordinates of the 66 features obtained through OpenCV and MediaPipe, we calculate various joint angles with the Equations (1) and (2) for Euclidean distances to determine body part lengths. Then, using trigonometric formulas, we calculate major joint angles [14]. Through this process, 12 joint data points are extracted and used as data.
(1)$$a=\sqrt{{\left(p1.x-p2.x\right)}^{2}+{\left(p1.y-p2.y\right)}^{2}{+\left(p1.z-p2.z\right)}^{2}}\;b=\sqrt{{\left(p2.x-p3.x\right)}^{2}+{\left(p2.y-p3.y\right)}^{2}{+\left(p2.z-p3.z\right)}^{2}}\;c=\sqrt{{\left(p3.x-p1.x\right)}^{2}+{\left(p3.y-p1.y\right)}^{2}{+(p3.z-p1.z)}^{2}}$$
(2)$$∠p1p2p3=\frac{180{}^{\circ}}{\pi }acos\left(\frac{a^{2}+b^{2}-c^{2}}{2ab}\right),\;p1,p2,p3\;:\;\;joint\;Point$$

#### 2.4.3. Selecting the Classification Model

To select an appropriate classification model for this study, we designed and comparatively analyzed various models. The first method identified the periodicity of joint angles through the power spectrum, setting this component as a key feature. We then experimented with machine-learning models and deep-learning models like LSTM and the CNN to choose the most effective model.

K-nearest neighbors (KNN), being a distance-based learning algorithm, requires the standardization of all training data. Figure 5 shows the graph of accuracies based on the number of K neighbors, and even at the highest K value, KNN showed objectively low performance (0.4632).

Random forest was experimented with by tuning hyperparameters with GridSearchCV [15]. It achieved an accuracy of about 0.72, which is an improvement from KNN. However, as seen in Table 1, it showed low accuracy in classifying the leg raise exercise. A closer look at Figure 6 is marked red box reveals that the leg raise was misclassified as either bicycle crunch or bird dog, both of which are exercises performed while on the floor, indicating the challenge in classification with random forest’s simple branching model.

We experimented with angle data based on coordinates detected with MediaPipe by using data from the frames at the start of a video as a baseline. To assess the baseline performance of LSTM and to prevent overfitting, dropout was incorporated.

As observed in Figure 7, with an increase in epochs, the loss decreased, and performance on the test data showed a slightly lower result (0.6721) compared to machine learning.

To achieve better results using LSTM directly, we utilized not only data from the original model but also standardized coordinate data from each video, incorporating the relative positions of joint coordinates.

From Figure 8, we observed that training with additional data along with the original data yielded a result higher than random forest (accuracy: 0.7950). We also experimented with combining a CNN at the front end of the previously used LSTM model and applied the optimal parameters found through hyperparameter tuning.

Figure 9 and Table 2 show that the new model demonstrated relatively good performance compared to the previous models yet still showed low accuracy in classifying the leg raise. Therefore, further refinement is necessary, such as developing a more complex model for classifying this exercise.

The experimental results of the above models are shown in Table 3. These models either analyzed the periodic features of each frame or applied LSTM to utilize the features of the entire dataset. However, such approaches are limited when predicting dynamic changes in each frame.

In this study, we overcome this limitation by developing a model based on features extracted from each video frame. We adopt a method that learns the exercise actions that change in each frame and selects the class with the highest probability predicted in each frame as the final output.

To implement this approach, we introduce a soft voting mechanism. Soft voting is an ensemble technique that probabilistically combines predictions from multiple models to make a final decision. By incorporating this mechanism, we can sum up the probabilities for each frame to reduce uncertainty and improve the overall classification performance.

Table 4 shows the CNN architecture used in this study. This model takes 78 attributes as input and output probabilities for 10 different exercises after passing the data through multiple layers.

After training the above model with video data from the Fitness Basic Dataset, we used it to output results on test data videos for ensemble learning.

#### 2.4.4. Select the Number of Frames

This study assists with a solution that automatically classifies exercises and provides real-time correction. Therefore, classification must be quick, so we find a range that has high performance from a minimal number of frames.

Figure 10 shows the data cross-validated using five stratified k-folds. Observing the accuracy per frame, the highest value was from about 70 frames, but, in reality, performance above 0.92 was reached after approximately 30 frames. Therefore, we decided to apply ensemble learning at 30 frames, where both quick judgment and high performance were achieved.

### 2.5. Real Data Collection and Preprocessing

In addition to the virtual dataset previously described, we collected real data to enhance the robustness of our findings. Sessions were recorded with people performing the 10 different exercises in our classification schema.

Video frames were preprocessed using MediaPipe to extract joint coordinates, which were then used to compute the same features found in the virtual dataset. This ensured consistency in the feature space between the virtual and real datasets.

## 3. Experimental Results

### 3.1. Classification Results from Virtual Data

The classification model developed in this study successfully demonstrated its ability to recognize and classify various exercise postures using Infinity AI’s Fitness Basic Dataset. Figure 11 depicts a confusion matrix showing the results of cross-validated test data. Among the 10 exercises in the dataset, the arm raise was recognized with near-perfect accuracy, and even the leg raise, which showed lower accuracy in other models, demonstrated higher accuracy in this model. Table 5 presents the evaluation metrics from the final model. Accuracy was 92.12%, indicating the system is highly effective at identifying various exercises. The F1 score was 91.58%, representing a balance between precision and recall, suggesting harmony between those two metrics. These results indicate that the proposed model provides reliable performance in real-time exercise classification and is highly suitable for real-world environments.

However, some exercises were not classified correctly. As seen in Figure 11, some instances of the bicycle crunch were classified as a leg raise, and the leg raise exercise was put into other classes. This could be due to the two exercises sharing similar features, or from the distinction between the two not being adequately learned from the training dataset. Additionally, the fact that the Superman exercise was classified as either a leg raise or a push-up suggests that similar angles and features in movements make it challenging to distinguish them clearly.

### 3.2. Classification Results from Real Data

We extended our analysis to include real data, yielding the results shown in Figure 12. Our system achieved an overall accuracy of 92.6282%, with specific exercise classifications detailed in the matrix. Notably, the curl exercise demonstrated a high classification rate, similar to the results from the virtual dataset. However, the arm raise exercise showed some misclassifications, possibly due to more variable executions in a real setting.

## 4. Discussion

This research validated a novel CNN ensemble model for accurate real-time exercise classification using video frames, demonstrated by its high performance on the Infinity AI’s Fitness Basic Dataset. Our model excelled in classifying diverse exercises, notably the arm raise, squats, and overhead press.

Compared to previous research, our proposed system confirmed various exercises by using an open dataset, and the optimized model provided high performance. Utilizing an open dataset allowed us to classify exercises from various angles and under various conditions. Additionally, using a CNN ensemble showed better performance than using LSTM alone. This improvement is likely due to the reduction in anomalies in each frame achieved by the ensemble.

Our system can be utilized for in-home training, rehabilitation programs, and other exercise programs that require posture correction. By classifying exercises in real time, the system can immediately provide users with appropriate corrective measures for each exercise, thereby enhancing the efficiency of the exercise and reducing the risk of injury.

The utilization of real data in this study, while enhancing the practical applicability of our model, introduced certain complexities. Real data, characterized by inherent noise and variability, demands extensive preprocessing and may introduce bias, potentially impacting model generalization. Additionally, ethical and privacy concerns over real data usage necessitate stringent anonymization and compliance with data protection laws, posing challenges to data accessibility and diversity. These factors underscore the necessity for advanced preprocessing techniques and the acquisition of varied datasets to ensure broader model applicability and accuracy in diverse real-world scenarios.

Real data enabled a direct comparison with the virtual dataset’s performance. Interestingly, classification accuracy for the curl remained high from both datasets, while the arm raise showed a decrease in precision. This may indicate that the virtual dataset, despite its advantages in controlled conditions and reproducibility, may not fully capture the nuances of real exercises.

The implications of these findings are multifaceted. The similarity in performance for exercises like the overhead press and the curl from both datasets suggests that our model can generalize certain exercises quite well. Conversely, the drop in performance for the arm raise and squat highlights the challenges faced when transferring knowledge acquisition from virtual to real data. This fact points to a need for further model refinement, potentially through advanced feature extraction that can capture the subtleties of human movement more effectively.

## 5. Conclusions

In this study, we proposed and validated an improved model that trains a CNN ensemble by using video frames for real-time exercise classification. This model trained with Infinity AI’s Fitness Basic Dataset proved capable of rapidly and accurately recognizing and classifying various exercises, achieving particularly high accuracy for the arm raise (0.98), squat (0.96), and overhead press (0.99).

Good performance from the model is evidenced by high levels of accuracy (92.12%), precision (91.62%), recall (91.64%), and F1 score (91.58%). The proposed model is expected to make a significant contribution to the development of real-time exercise classification and corrective solutions. The results from this study suggest the potential for broader applications of the model, especially in personalized fitness recommendations and physical therapy services.

However, despite the success, we observed challenges in classifying certain exercises with similar movements or features, such as the bicycle crunch and the leg raise, which contributed to 7.8% of exercises not being accurately classified. These instances highlight the complexity of exercise movements and the need for more sophisticated modeling techniques to improve classification accuracy. Future research should focus on enhancing the model’s ability to differentiate between exercises with subtle variations in movements. This may involve incorporating additional data sources, such as temporal movements or deeper feature extraction methods, to capture the nuances of each exercise more effectively. Additionally, expanding the dataset to include a broader array of exercises and variations could provide the model with a more comprehensive training base, further refining its predictive capabilities.

To address these challenges and push the boundaries of real-time exercise classification, the continued exploration and development of complex models are essential. Efforts should be directed towards not only improving the model’s accuracy but also ensuring its adaptability and effectiveness in real-world fitness and therapeutic contexts. By achieving these goals, we can enhance the utility of technology-driven fitness recommendations and support more personalized, effective physical therapy interventions.

## Figures and Tables

**Figure 1 sensors-24-03133-f001:**
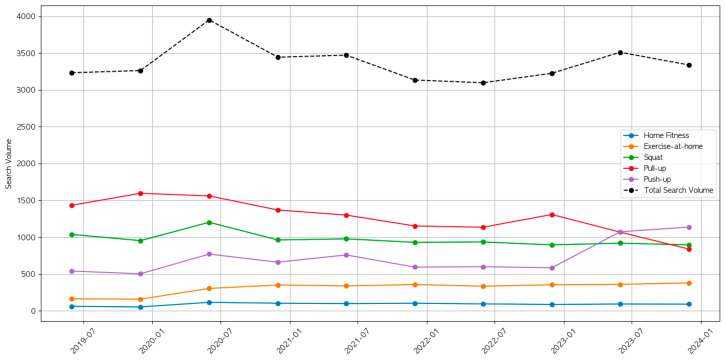
Home training terms searched for on Google.

**Figure 2 sensors-24-03133-f002:**
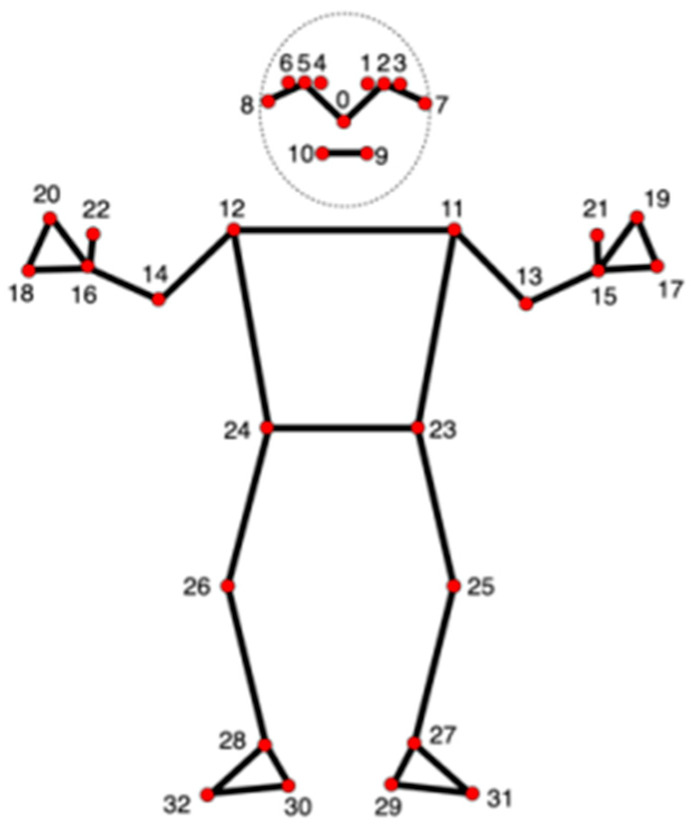
MediaPipe pose landmarks.

**Figure 3 sensors-24-03133-f003:**
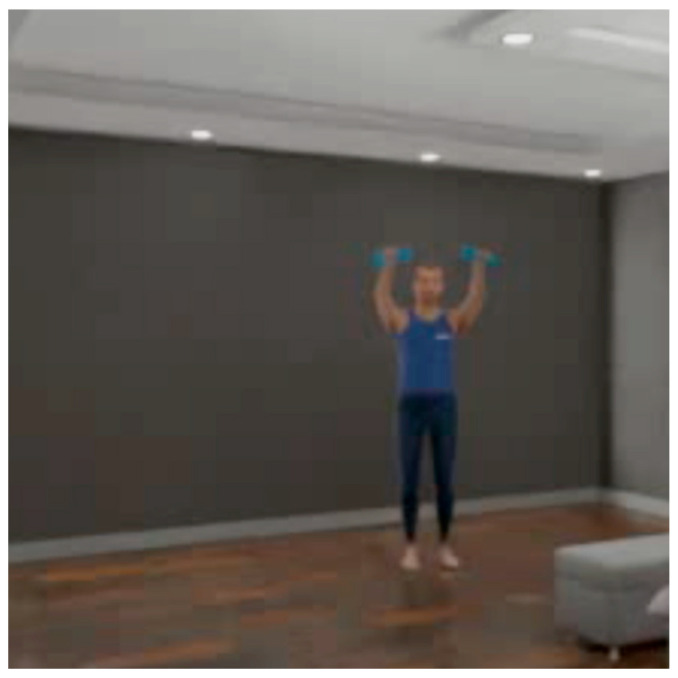
InfiniteRep video screenshot.

**Figure 4 sensors-24-03133-f004:**
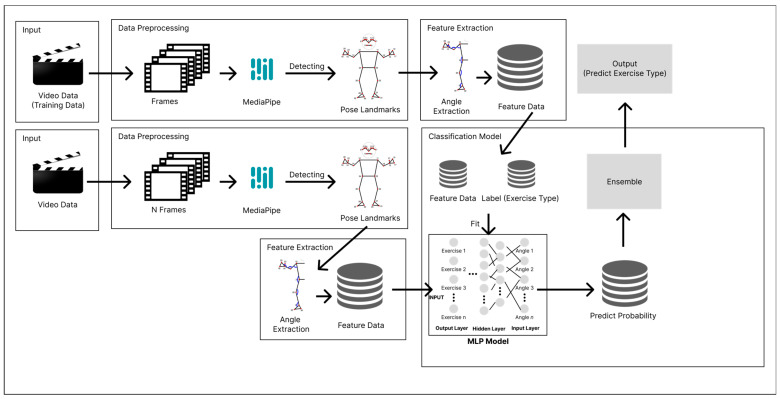
The system architecture.

**Figure 5 sensors-24-03133-f005:**
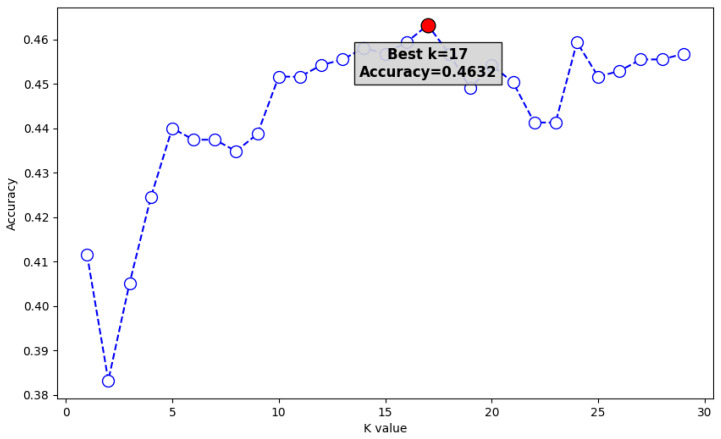
KNN accuracy vs. K values.

**Figure 6 sensors-24-03133-f006:**
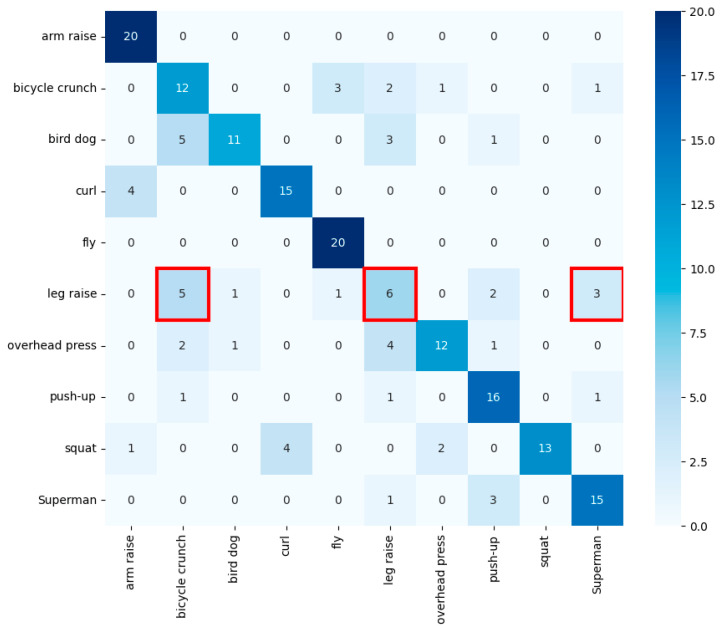
Random forest’s confusion matrix. The red box indicates frequent misclassifications of leg raises as bicycle crunches or bird dog.

**Figure 7 sensors-24-03133-f007:**
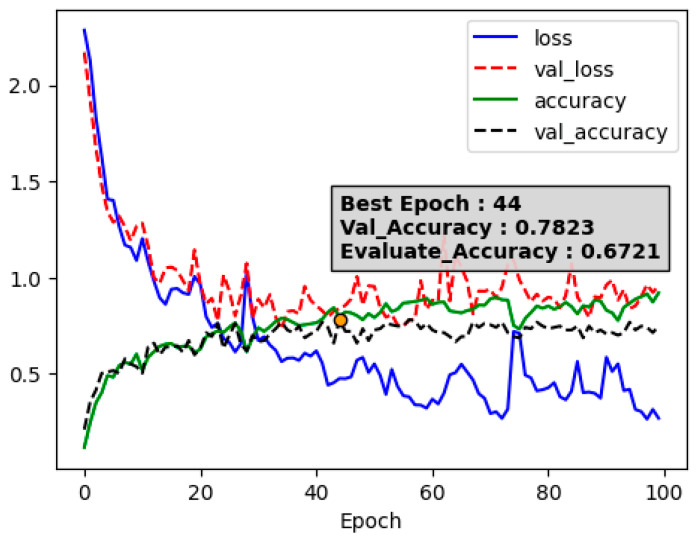
LSTM loss and accuracy.

**Figure 8 sensors-24-03133-f008:**
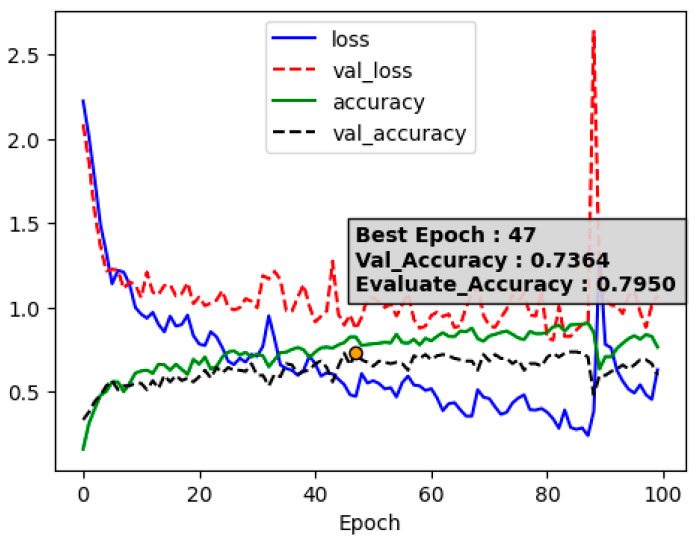
LSTM and accuracy (added data).

**Figure 9 sensors-24-03133-f009:**
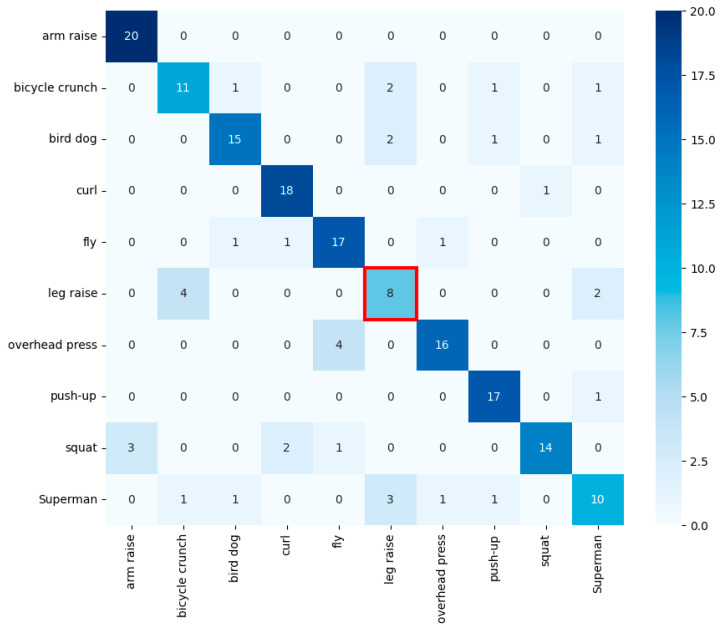
The CNN and LSTM confusion matrix. The red box highlights the cells where the leg raise exercise was classified, demonstrating the relatively lower accuracy in identifying this specific exercise compared to others.

**Figure 10 sensors-24-03133-f010:**
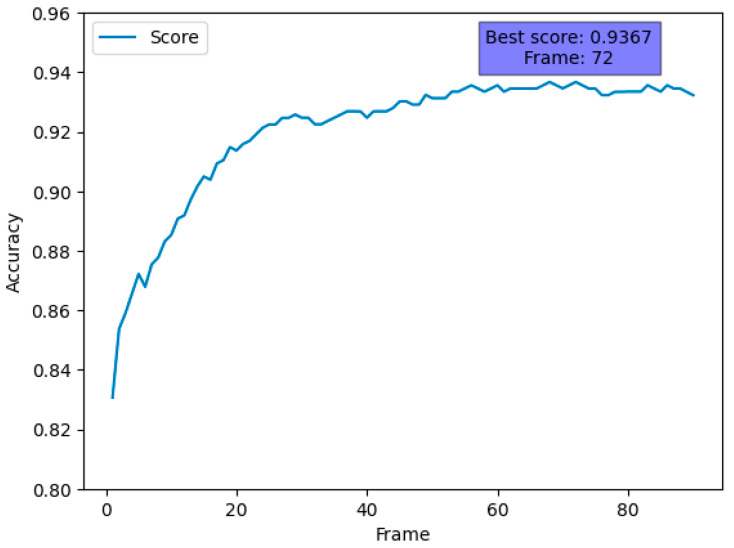
Accuracy per frame.

**Figure 11 sensors-24-03133-f011:**
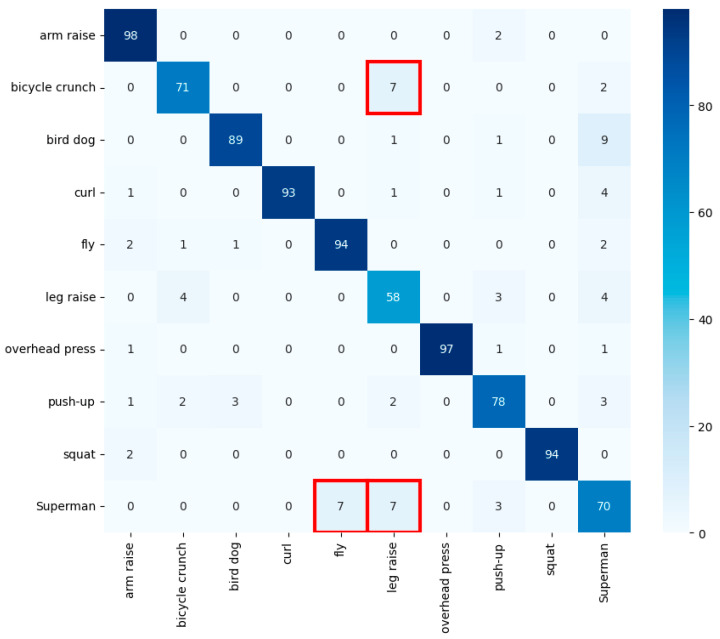
Confusion matrix for cross-validated test data. Red box indicates misclassifications among similar exercises, such as bicycle crunches, leg raises, and Superman, due to overlapping features.

**Figure 12 sensors-24-03133-f012:**
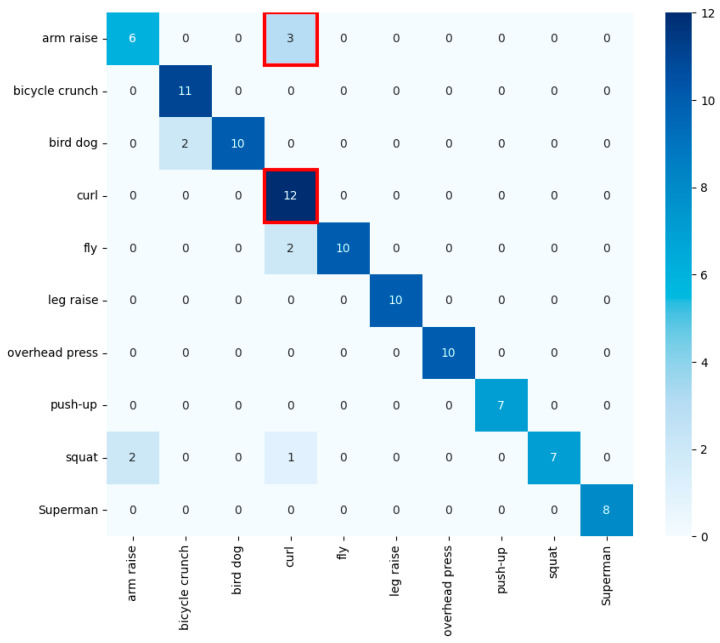
Confusion matrix from real data. red box indicates curls were classified accurately and arm raises exhibited notable misclassifications.

**Table 1 sensors-24-03133-t001:** Random forest’s classification metrics.

Class	Precision	Recall	F1 Score
arm raise	0.83	0.95	0.88
bicycle crunch	0.59	0.68	0.63
bird dog	0.69	0.55	0.61
curl	0.81	0.68	0.74
fly	0.86	0.95	0.90
** leg raise **	** 0.50 **	** 0.56 **	** 0.53 **
overhead press	0.71	0.50	0.59
push-up	0.70	0.84	0.76
squat	0.83	0.75	0.79
Superman	0.75	0.79	0.77

Entries highlighted in bold and red indicate the lowest accuracy rates or the highest misclassification rates for each exercise.

**Table 2 sensors-24-03133-t002:** CNN and LSTM classification metrics.

Class	Precision	Recall	F1 Score
arm raise	0.87	1.00	0.93
bicycle crunch	0.69	0.69	0.69
bird dog	0.83	0.79	0.81
curl	0.86	0.95	0.90
fly	0.77	0.85	0.81
** leg raise **	** 0.53 **	** 0.57 **	** 0.55 **
overhead press	0.89	0.80	0.84
push-up	0.85	0.94	0.89
squat	0.93	0.70	0.80
Superman	0.67	0.59	0.62

Entries highlighted in bold and red indicate the lowest accuracy rates or the highest misclassification rates for each exercise.

**Table 3 sensors-24-03133-t003:** Accuracy by model.

Model	Accuracy
KNN (k = 17)	0.4632
Random Forest	0.7268
LSTM	0.7950
CNN + LSTM	0.7978

**Table 4 sensors-24-03133-t004:** The CNN architecture.

Layer	Type	Maps	Size	Kernel Size
In	Input	1	78	-
C1	Conv1D	32	76	3
S2	MaxPooling1D	-	38	2
C3	Conv1D	64	36	3
S4	MaxPooling1D	-	18	2
C5	Conv1D	128	16	3
S6	MaxPooling1D	-	8	2
C7	Conv1D	256	6	3
S8	MaxPooling1D	-	3	2
F10	Flatten	-	768	-
F11	Fully connected	512	-	-
Out	Fully connected	10	-	-

**Table 5 sensors-24-03133-t005:** Evaluation metrics.

Metric	Value
Accuracy	0.9212
Precision	0.9162
Recall	0.9164
F1 Score	0.9158

## Data Availability

Data are contained within the article.

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
