# Peer review of "Workout Classification Using a Convolutional Neural Network in Ensemble Learning"

_sensors, 2024, doi:10.3390/s24103133_

Round 1

Reviewer 1 Report

Comments and Suggestions for Authors

This paper propose a new approach to real-time exercise posture classification tasks.The logic of the article is relatively clear and sufficient experiments have been conducted. My comments are as followes:

1.  The writing of the article is a bit redundant, especially the contribution part of the article. It is recommended to modify it to make it more concise.

2.  The method proposed by the author is not clearly marked in the experiment. It is recommended to highlight your experimental results in the table.

3.  There are few CNN methods compared by the author. Here are several reference articles that can be used for comparison:

a.  Exploring a fine-grained multiscale method for cross-modal remote sensing image retrieval

b.  Resnext and res2net structures for speaker verification

c.  Speech Emotion Recognition Based on Secondary Feature Reconstruction

4.  It is recommended to add more comparative experiments to prove the effectiveness of the method.

Comments on the Quality of English Language

good

Reviewer 2 Report

Comments and Suggestions for Authors

The following aspects must be updated:

-section 2 must be focused on existing methods, obtained results, datasets used for evaluation

-a separate section must justify the selection methods used for evaluation in this paper; also explain the selection of this dataset for the evaluation

-the discussion section must be extended with a set of comments regarding the possible disadvantages of using real data

-an ablation study is needed: compare these methods with other existing ones; also, the evaluation must be done using more datasets (also real data) and not only one dataset

-add a detailed description of the dataset; line 103: integrate in a sentence see figure 3

-do not start a section with a figure (eg. figure 1): first add some description about the figure and next place the figure

Round 2

Reviewer 2 Report

Comments and Suggestions for Authors

Part of my comments were addressed but still, there is a main part that must be extended. In order to prove the results of the proposed method an ablation study is needed: compare it with other existing ones; also, the evaluation must be done using more datasets (also real data) and not only one single dataset.
